# Exploring the facilitators, barriers, and strategies for self-management in adults living with severe mental illness, with and without long-term conditions: A qualitative evidence synthesis

Abisola Balogun-Katung[1,2], Claire Carswell[1]*, Jennifer V. E. Brown[1], Peter Coventry[1], Ramzi Ajjan[3], Sarah Alderson[4], Sue Bellass[4], Jan R. Boehnke[1,5], Richard Holt[6,7], Rowena Jacobs[8], Ian Kellar[9], Charlotte Kitchen[1], Jennie Lister[1], Emily Peckham[1], David Shiers[10], Najma Siddiqi[1,2], Judy Wright[4], Ben Young[1,11], Jo Taylor[1], on behalf of the DIAMONDS research team¶

1 Department of Health Sciences, University of York, York, United Kingdom, 2 Hull York Medical School, York, United Kingdom, 3 School of Medicine, University of Leeds, Leeds, United Kingdom, 4 Leeds Institute of Health Sciences, University of Leeds, Leeds, United Kingdom, 5 School of Health Sciences, University of Dundee, Dundee, United Kingdom, 6 Human Development and Health, Faculty of Medicine, University of Southampton, Southampton, United Kingdom, 7 Southampton National Institute for Health Research Biomedical Research Centre, University Hospital Southampton NHS Foundation Trust, Southampton, United Kingdom, 8 Centre for Health Economics, University of York, York, United Kingdom, 9 School of Psychology, University of Leeds, Leeds, United Kingdom, 10 Psychosis Research Unit, Greater Manchester Mental Health NHS Foundation Trust, Manchester, United Kingdom, 11 Institute of Health and Wellbeing, University of Glasgow, Glasgow, United Kingdom

¶ Membership of the DIAMONDS research team is provided in the Acknowledgments.
* Claire.carswell@york.ac.uk

**Data Availability Statement:** The manuscript is a qualitative evidence synthesis, therefore the data

## Abstract

### Background

People living with severe mental illness (SMI) have a reduced life expectancy by around 15–20 years, in part due to higher rates of long-term conditions (LTCs) such as diabetes and heart disease. Evidence suggests that people with SMI experience difficulties managing their physical health. Little is known, however, about the barriers, facilitators and strategies for self-management of LTCs for people with SMI.

### Aim

To systematically review and synthesise the qualitative evidence exploring facilitators, barriers and strategies for self-management of physical health in adults with SMI, both with and without long-term conditions.

### Methods

CINAHL, Conference Proceedings Citation Index- Science, HMIC, Medline, NICE Evidence and PsycInfo were searched to identify qualitative studies that explored barriers, facilitators

used in the synthesis are available through the References.

**Funding:** This paper reports work undertaken as part of the DIAMONDS programme, which is funded by the National Institute for Health Research under its Programme Grants for Applied Research (project number RP-PG-1016-20003). Peter Coventry is part funded by the UK Research and Innovation Closing the Gap Network+ [ES/S004459/1].

**Competing interests:** The authors have declared that no competing interests exist.

and strategies for self-management in adults with SMI (with or without co-morbid LTCs). Articles were screened independently by two independent reviewers. Eligible studies were purposively sampled for synthesis according to the richness and relevance of data, and thematically synthesised.

## Results

Seventy-four articles met the inclusion criteria for the review; 25 articles, reporting findings from 21 studies, were included in the synthesis. Seven studies focused on co-morbid LTC self-management for people with SMI, with the remaining articles exploring self-management in general. Six analytic themes and 28 sub-themes were identified from the synthesis. The themes included: the burden of SMI; living with co-morbidities; beliefs and attitudes about self-management; support from others for self-management; social and environmental factors; and routine, structure and planning.

## Conclusions

The synthesis identified a range of barriers and facilitators to self-management, including the burden of living with SMI, social support, attitudes towards self-management and access to resources. To adequately support people with SMI with co-morbid LTCs, healthcare professionals need to account for how barriers and facilitators to self-management are influenced by SMI, and meet the unique needs of this population.

## Introduction

Severe mental illnesses (SMI) such as schizophrenia and bipolar disorder affect around 1% of the population [1] and are associated with a reduced life expectancy by around 15–20 years compared with the general population [2]. This is mostly explained by poorer physical health including higher rates of non-communicable long-term conditions (LTCs) such as diabetes and heart disease, and worse self-management of those conditions [3]. Adequately managing LTCs necessitates engaging in daily self-management, such as taking medications and reducing risks through stopping smoking, eating healthily, and being physically active. Education and support programmes aimed at increasing people's knowledge, skills, and confidence to manage their condition in their daily lives and reduce the risk of complications are key elements of care for people with LTCs. Building on considerable evidence about challenges to self-management, these programmes are widely understood to be effective for the general population without SMI [4].

Far less is known about the challenges to LTC self-management for people with SMI. A recent survey of people with SMI and co-morbid diabetes in England reported that people with SMI engage in less diabetes self-management than those without SMI [5]. It was found that they had lower levels of healthy eating, physical activity and monitoring of symptoms and complications, but similar medication-taking behaviours [5]. Another study that focused specifically on medication adherence in people with Type 2 diabetes in the US found that people with schizophrenia were more likely to adhere to hypoglycaemic medication than those without [6].

Existing literature alludes to some of the reasons why people with SMI may struggle more with self-management. For example, SMI is characterised by disturbances of thought,

perception, emotional expression and motivation [7], which may influence self-efficacy, literacy, lifestyle, and behaviour [8, 9]. The physical health of people with SMI may also be overlooked as their mental illness is prioritised, and physical health symptoms may be attributed to the underlying mental illness; a an example of diagnostic overshadowing [10]. Experiences of treatment for SMI may also influence how people view their role in managing their health, affecting, for example, perceived control and involvement in decision-making [11, 12]. Capacity and confidence to self-manage might also be negatively affected by stigma associated with mental illness and discrimination [13]. Additionally, people with SMI are more likely to experience financial hardship, housing insecurity or social isolation [14, 15], making it more difficult to make healthy lifestyle choices and access healthcare services and interventions [16].

As a consequence of these many barriers, people with SMI might find it difficult to effectively engage with physical health self-management programmes designed for people without SMI. People with SMI also tend to be excluded from trials assessing effectiveness of these programmes [17]. This points to the need for more tailored interventions that target the challenges people with SMI experience in relation to their self-management [18]. To support the development of these interventions, it is imperative to first understand the way the lived experience of SMI influences people's engagement with self-management.

We therefore systematically reviewed and synthesised qualitative evidence about the experiences of self-management in people with SMI, both with and without LTCs, to understand the barriers, facilitators, and strategies for self-management of physical health in this population.

## Materials and methods

This systematic review and qualitative synthesis is part of the DIAMONDS research programme, and informs the development of an evidence-based intervention to support self-management of diabetes in people with SMI [19]. The protocol was prospectively registered on PROSPERO (CRD42018099553). Here we report findings from the qualitative studies in the review. Findings from quantitative studies have been reported [20].

Protocol amendments include a decision to purposively sample studies most likely to be of utility for the synthesis based on data richness and data saturation and use of inductive thematic analysis as the original planned framework was found to be too restrictive (see S1 Appendix) for details of all changes).

### Eligibility criteria

Qualitative studies of any design that explored the barriers, facilitators and strategies for self-management in adults with SMI (with or without co-morbid LTCs) were eligible for inclusion, from the perspective of both adults with SMI within the community, and people who provide support for adults with SMI, such as healthcare professionals. In studies where a specific psychiatric diagnosis was not named, we took the label 'SMI' to indicate the presence of a serious mental illness. We defined self-management as "all the actions taken by people to recognise, treat and manage their own healthcare independently of or in partnership with the healthcare system" [21], and adopted the American Association of Diabetes Educator's self-care behaviours (AADE-7) as a framework to determine what actions constitute self-management [22]. Studies were only included if they were published in the English language and conducted in a high income (OECD member) country [23], to ensure experiences were representative of similar healthcare systems.

The full criteria for inclusion are summarised in Table 1.

**Table 1. Inclusion criteria for qualitative synthesis.**

| | Inclusion criteria |
|---|---|
| **Study population** | Adults aged 18 or over |
| | *(for mixed populations at least 70% were aged 18 or over)* |
| | Diagnosed with SMI which includes schizophrenia, affective disorders (psychotic), bipolar disorder, paranoid disorders or psychosis |
| | *(we included mixed studies when the study was about people with severe and enduring mental illness, but which also included conditions without psychosis, e.g. major depression, personality disorder)* |
| **Study focus** | Studies had to explore barriers, facilitators and strategies for self-management. |
| **Study design** | Qualitative studies which were defined as studies that collected data using specific qualitative techniques such as unstructured interviews, semi-structured interviews or focus groups, either as a stand-alone methodology or as discrete part of a larger mixed-method study, and analysed qualitatively. Studies that collected data using qualitative methods but then analysed these data using quantitative methods were excluded. |
| **Study participants** | People with SMI and/or those who provide care or support to people with SMI (e.g. informal carers, health and social care staff) |
| **Study setting** | Community settings (e.g. people with SMI could be living at home or in long-term residential settings) |
| | *(for mixed populations at least 70% were living in a community setting)* |
| **Study country** | High income countries only (i.e. those with similar healthcare systems), defined as OECD member countries [23]. |
| **Article type** | Articles published in peer-reviewed journals |
| **Publication language** | English language only |
| **Publication date** | No restriction |

## Search strategy

Electronic databases; Ovid MEDLINE(R) and Epub Ahead of Print, In-Process & Other Non-Indexed Citations and Daily 1946+, CINAHL (EBSCOhost), PsycINFO (Ovid) 1806+, Conference Proceedings Citation Index–Science (Clarivate Analytics) 1990+, HMIC Health Management Information Consortium (Ovid), and NICE Evidence Search were searched on 25th July 2018. No limits were placed on date of publication. Update searches were performed in MEDLINE and PsycINFO on 21st November 2019 and 27th August 2020, as these were the two databases that generated the most eligible studies. Reference lists of relevant systematic reviews and included studies were also searched.

Searches were developed for the following concepts: severe mental illness, self-management, healthy lifestyle, and barriers/motivators; and peer-reviewed by an Information Specialist. Published search strategies were used for the SMI concept [24] and self-management concept [25] with minor adaptations. No date or language limits were applied to the searches. Commentaries, letters, and editorials were removed from the update searches as these were unlikely to provide full study data. (see S2 Appendix for MEDLINE Ovid search strategy). Original and update search results were stored and deduplicated in an EndNote library following the AUHE Duplicates Guide [26] to remove high certainty duplicates automatically and check low certainty duplicates manually.

## Study selection

De-duplicated search results were assessed independently by two reviewers in Covidence [27], with ineligible citations first excluded by title and abstract. Full-text articles of the remaining

results were retrieved and assessed for eligibility. Disagreements were resolved via a third reviewer. Reasons for exclusion of full-text articles were recorded.

## Data extraction

Study and participant characteristics, study methods and focus were extracted into Microsoft Excel using a piloted data extraction template, shown in Table 2. The results of included studies, including author-reported results, direct quotations, and results tables, were imported into NVivo version 12 for synthesis [28].

## Data relevance

We used the data richness scale developed by Ames et al. (2017) [29] to score each study according to data richness (as a measure of quality) and relevance (to the review aim) and used this as a key element of our purposive sampling strategy for selecting studies to include in the thematic synthesis [29–31]. Two reviewers independently scored each eligible study, with disagreements resolved via a third reviewer (see S3 Appendix for scoring criteria).

## Purposive sampling strategy

We applied a two-stage strategy to select a purposive sample of studies to include in the thematic synthesis, to manage the amount of data identified during screening and ensure the most relevant and rich data was included in the synthesis [29, 31]:

1. Inclusion of all papers scoring 4 or 5 on the data richness scale and exclusion of all studies scoring 1 or 2

2. Selection of a sample of studies scoring 3 based on representation of a range of comorbid LTCs and SMIs, and exploration of different AADE-7 self-management behaviours. We prioritised studies which included people with SMI as participants to better understand their perspectives.

During the synthesis we also monitored data saturation and continued to add other studies scoring 3 as needed until we were satisfied that data saturation had been achieved [32, 33].

## Thematic synthesis

We followed the three stages of thematic synthesis [34], adapting the process to meet the requirements of our review:

1. 'Free coding'

One reviewer coded the extracted results of included studies in NVivo, with codes labelled according to the underlying meaning and content of the text being coded. We adopted a pragmatic approach in which results that were not relevant to the review aim were not coded. Additionally, rather than forcing a meaning on individual sentences, we coded segments of text that contained unique content. The coding process was reviewed regularly with a second reviewer to assess the translation of codes from one study to another, and ensure each code reflected a unique idea or concept.

2. Organising 'free codes' into descriptive themes

Similar codes were grouped to identify descriptive themes that represented the findings across studies, with coded data re-examined to develop our understanding of each theme and mind maps used to explore potential relationships between codes and themes.

3. Developing analytical themes

**Table 2. Characteristics of studies included in the thematic synthesis.**

| Data richness score | Study First author, year Country Setting | Study sample Mean age Sample size % female Ethnicity Patients/HCPs | Type of SMI (%) Mean duration | Type of LTC (%) Mean duration | Study aims | Reported self-management behaviours* | Methodology Data collection method |
|---|---|---|---|---|---|---|---|
| | | | | STUDIES ABOUT PEOPLE WITH SMI AND CO-MORBID LTC | | | |
| 5 | Blixen, 2016a [35] USA Urban 'safety net' care service | 53.9 years N = 20 70% African-American (50%) Caucasian (40%) Hispanic (10%) Patients | Schizophrenia or schizoaffective disorder (4%) BD (15%) Major depression (65%) 19.7 years | DM (type not specified; 100%) 10 years | Assess perceived barriers to self-management among patients with SMI and DM. | Healthy eating Being active Monitoring Taking medication Healthy coping | Phenomenology Interviews |
| 5 | Cimo, 2018 [36] Canada Community based | Age NR N = 17 % female NR Ethnicity NR Patients | Schizophrenia (6%) BD (17%) Depressive disorder (53%) Severe anxiety (6%) Multiple diagnoses (18%) Duration NR | Inclusion criteria: DM/pre-DM 'borderline DM' 'borderline high sugars' 'slightly high blood sugar levels' Duration NR | Explore patients' perspectives of their challenges engaging with DM self-care behaviours. | Healthy eating Monitoring Taking medication Problem solving Reducing risks | Thematic analysis Focus groups |
| 5 | El-Mallakh, 2006 [37] (reported in El-Mallakh 2006 [37] and El-Mallakh 2007 [38]) USA Community MH centre | 50.3 years N = 11 45% European American (45%) African American (55%) Patients | Schizophrenia (91%) Schizoaffective disorder (9%) Duration NR | Inclusion criteria: T1DM T2DM Duration NR | Develop a theory of self-care for individuals with comorbid schizophrenia/schizoaffective disorder and DM. Examine the approaches of MH consumers with comorbid schizophrenia/schizoaffective disorder and DM to diabetic self-care. | Healthy eating Being active Monitoring Taking medication Reducing risks | Grounded theory/ constant comparison method Interviews |
| 5 | Kuyahnytska, 2018 [39] Canada Urban family health setting | Patient participants: Range 40–66 years N = 10 50% Caucasian (50%) 'visible minority' (50%) HCP participants: Age NR N = 5 80% Caucasian (100%) | Schizophrenia (70%) BD (30%) Duration NR | T2DM (100%) Duration NR | Explore everyday experiences of DM self-management by people diagnosed with SMI. | Healthy eating Being active Taking medication | Critical ethnography Interviews |
| 5 | Mulligan, 2017 [40] UK Inner city community MH service | 47 years N = 14 36% Black African Caribbean (36%) South Asian (36%) White (7%) Mixed White/ African Caribbean (21%) Patients | Schizophrenia (50%) Schizoaffective disorder (7%) BD (21%) Personality disorder (7%) Depression with psychotic features (14%) Median 7 years | T2DM (100%) Median 6 years | Identify barriers and enablers to effective DM self-management experienced by people with SMI and T2DM. | Healthy eating Being active Monitoring Taking medication Reducing risks Healthy coping | Analysis informed by theoretical domains framework Interviews |
| 4 | Blixen, 2018 [41] USA Urban academic medical centre | 52.8 years N = 13 79.6% Caucasian (7.7%) African American (92.3%) Patients | BD (100%) Age of onset of BD: 28.77 | Hypertension (100%) | To obtain information from patients with both BD and hypertension that would inform the development of m-Health intervention to improve medication adherence for poorly adherent individuals living with both conditions. | Taking medication Problem solving Healthy coping | Content analysis with an emphasis on dominant themes Focus groups |

*(Continued)*

**Table 2.** (Continued)

| Data richness score | Study First author, year Country Setting | Study sample Mean age Sample size % female Ethnicity Patients/HCPs | Type of SMI (%) Mean duration | Type of LTC (%) Mean duration | Study aims | Reported self-management behaviours[a] | Methodology Data collection method |
|---|---|---|---|---|---|---|---|
| 4 | Stenov, 2020 [42] Denmark Regional psychiatry outpatient clinics | 47 years N = 15 40% Ethnicity NR Patients | Schizophrenia (60%) Schizoaffective disorder (7%) Bipolar disorder (13%) Personality disorder (7%) Severe depression (3%) Duration NR | T1DM (20%) T2DM (80%) | Gain insight into life with co-existing DM and SMI to identify the challenges specific to this condition and support needs for diabetes care | Healthy eating Being active Monitoring Reducing risks Healthy coping | Systematic text condensation Interviews |
| | | | | | STUDIES ABOUT PEOPLE WITH SMI | | |
| 5 | Jimenez, 2017 [43] (reported in Jimenez 2017 [43], Jimenez 2016 [44], and Jimenez 2015 [45]) USA Community MH centre | 40.3 years N = 20 45% Latino (100%) Patients | Schizoaffective disorder (50%) Schizophrenia (25%) Severe major depressive disorder (15%) BD (10%) Duration NR | NR | To identify the role of SMI in motivation, participation and adoption of health behaviour change. | Healthy eating Being active Reducing risks | Thematic analysis Interviews |
| 4 | Blixen, 2016b [46] USA Urban hospital | 47.29 years N = 21 71.4% African American (61.9%) Caucasian (23.8%) Hispanic (4.8%) Other (14.3%) Patients | BD Type I (81%) BD Type 2 (14.3%) Duration NR (mean age at onset 22.05 years) | Reported from larger RCT sample: Hypertension (45.7%) Arthritis (45.6%) High cholesterol (38.0%) | To explore patients' perceptions of barriers to self-management of BD. | Taking medication Healthy coping | Thematic analysis Interviews |
| 4 | Chee, 2019 [47] Australia Community treatment | 26 years N = 24 8% Ethnicity NR Patients | Psychosis (100%) Duration NR | NR | To explore young mental health consumers' level of knowledge and understanding of the impact their psychosis had on their overall health and well-being and their physical health needs. | Healthy eating Being active Taking medication Reducing risks Healthy coping | Grounded theory Interviews |
| 4 | Johnstone, 2009 [48] UK Community MH teams | 43 years N = 27 40.7% Ethnicity NR Patients | Schizophrenia (100%) Duration NR | NR | To investigate the barriers to uptake of and adherence to physical activity in community-dwelling patients diagnosed with schizophrenia. | Being active | Interpretive phenomenological analysis Interviews |
| 4 | Nakanishi, 2019 [49] Japan HeAL Japan community workshops | 30–50 years N = 37 0% NR Patients and HCPs | Schizophrenia (100%) Duration NR | NR | To clarify the critical mechanism underlying autonomy in physical health promotion based on the perspectives of people with severe mental illness. | Healthy eating Taking medication Healthy coping | Content analysis Panel discussions during workshop |
| 4 | Rastad, 2014 [50] Sweden Outpatient clinics | Range 22–63 years N = 21 35% Ethnicity NR Patients | Schizophrenia (90%) Schizoaffective disorder (10%) Duration NR | NR | To study the perception and experience of barriers to and incentives for physical activity in daily living in patients with schizophrenia. | Being active Problem solving Reducing risks | Conventional qualitative content analysis Interviews |
| 4 | Shor, 2013 [51] Israel Community residential MH facilities | 36.27 years N = 84 49% Ethnicity NR Patients | SMI not specified, participants were recruited from health promotion groups where the criteria for participation included: diagnosis of a long and persistent mental illness; taking antipsychotic medications; and meeting at least two of the follow criteria: overweight, difficulties maintaining health nutrition habits, or not physically active. Duration NR | Sixty-seven percent of the participants reported that they have physical problems in addition to the mental illness. | To examine the perceived barriers affecting the ability of persons with SMI from incorporating healthy nutritional practices and physical activities in their lives. | Healthy eating Being active | Grounded theory Interviews |

(*Continued*)

**Table 2.** (Continued)

| Data richness score | Study First author, year Country Setting | Study sample Mean age Sample size % female Ethnicity Patients/HCPs | Type of SMI (%) Mean duration | Type of LTC (%) Mean duration | Study aims | Reported self-management behaviours[a] | Methodology Data collection method |
|---|---|---|---|---|---|---|---|
| 4 | Wheeler, 2018 [52] Australia Community mental health support provided by NGOs | 38.2 years N = 20 50% Ethnicity NR Patients/Exercise practitioners | Schizophrenia (36%) Bipolar disorder (21%) Depression (7%) Agoraphobia (7%) Multiple diagnosis (29%) | NR | To better understand the determinants of engagement in exercise for consumers experiencing mental health problems. | Being active | Interpretive phenomenological analysis Interviews |
| 3 | Barre, 2011 [53] (reported in Barre 2011 and Glover 2013) USA Outpatient MH centre | Range 30–61 years N = 31 51.6% Caucasian (54.8%) African American (35.5%) Patients | Schizophrenia/schizoaffective disorder (35.5%) BD (35.5%) Major depression (29%) Duration NR | NR | To explore understanding of a healthy diet and the barriers to healthy eating in persons with serious mental illnesses. To document, analyse and understand self-identified barriers to exercise specific to people living with serious mental illnesses. | Healthy eating Being active | Thematic analysis Interviews |
| 3 | Heffner, 2018 [54] USA Community based | 49 years N = 10 80% Caucasian (100%) Patients | BD type 1 (50%) BD type 2 (50%) Duration NR | NR | Explore challenges and facilitators of quitting for smokers with BD. | Healthy coping | Inductive content analysis Interviews |
| 3 | Keller-Hamilton, 2019 [55] USA Community MH clinic | 46 years N = 24 62.5% Ethnicity NR Patients | *Inclusion criteria:* Schizophrenia Schizoaffective disorder Bipolar disorder with psychotic features Psychosis not otherwise specified Duration NR | NR | To report reasons for smoking and barriers to cessation that are both related and unrelated to SMI symptoms among adults with SMI. | Healthy coping | Thematic analysis Focus groups |
| 3 | Pearsall, 2014 [56] UK Community MH setting | 54.6 years N = 13 50% Ethnicity NR Patients | *Inclusion criteria:* Schizophrenia Schizoaffective disorder Bipolar affective disorder Durations NR | NR | To understand the problems experienced by individuals with SMI when asked to attend a healthy living program. | Healthy eating Being active Reducing risks Healthy coping | Grounded theory/ thematic analysis Interviews |
| 3 | Wardig, 2013 [57] Sweden Outpatient psychiatric facilities | Median 46 years N = 40 47.5% Ethnicity NR Patients | Schizophrenia (33%) Schizoaffective disorder (33%) BD (20%) Delusional disorder (7%) Unspecified psychosis (7%) Range 1–40 years | NR | To explore prerequisites for a healthy lifestyle as described by individuals diagnosed with psychosis. | Healthy eating Being active Healthy coping | Conventional content analysis Interviews |
| 3 | Williams, 2013 [58] Australia Community Street Soccer programme | Range 18–23 years N = 6 0% Ethnicity NR Patients | Schizophrenia (67%) Psychosis (17%) BD (17%) Duration NR | NR | To identify why young people who had experienced psychosis consistently decided to attend the street soccer programme. | Being active | Thematic analysis Interviews |

Abbreviations: BD–bipolar disorder; COPD–chronic obstructive pulmonary disorder; DM–diabetes mellitus; DRS—Data Richness Score (score 1–5 based on Ames et al 2017 scale); HCP–health care professional; LTC–long-term condition (physical); MH–mental health; N/A–not applicable; NR–not reported; PTSD–post-traumatic stress disorder; SMI–severe mental illness; T1DM–type 1 diabetes mellitus; T2DM–type 2 diabetes mellitus.

We interpreted what the descriptive themes inferred about the experiences of self-management, going beyond the findings of the primary studies to generate additional concepts and understandings.

## Results

Our searches identified 10,224 unique records, 9,832 of which were assessed as not relevant on title and abstract screening. Of the remaining 392 reviewed as full-text articles, 74 articles reporting the findings of 68 studies met the inclusion criteria for the review. Fig 1 shows the screening and selection process, and S4 Appendix provides a summary of the 68 studies meeting eligibility criteria for inclusion.

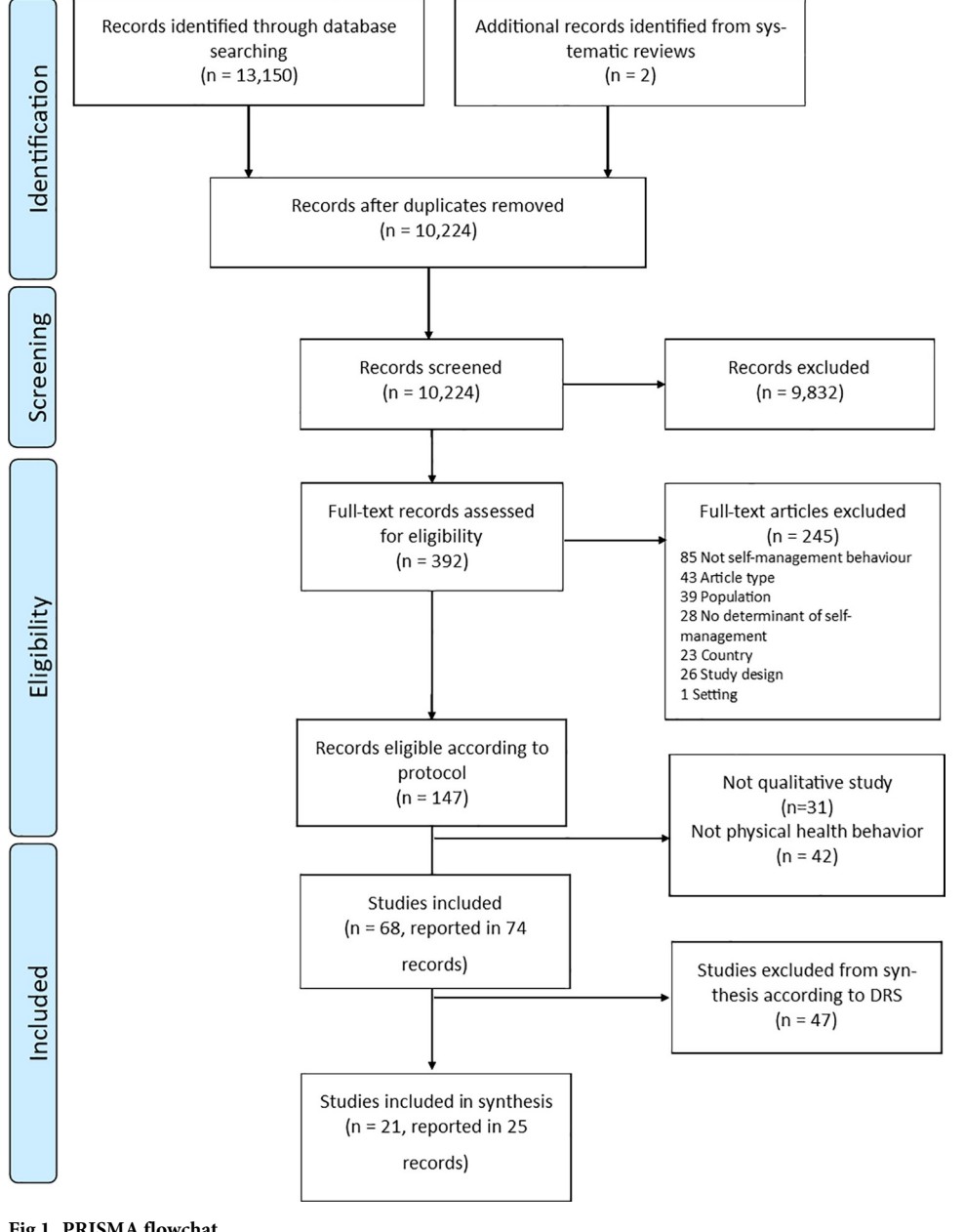

**Fig 1. PRISMA flowchat.**

## Appendix 4: S4 Appendix

Of the 68 eligible studies, 15 were given a data richness score of 4 (n = 9) or 5 (n = 6) and were therefore included in the qualitative synthesis. A further six studies scoring 3 were added to the synthesis, resulting in 21 studies (25 articles) included in the synthesis. Table 2 provides details about the characteristics and participants of studies included in the thematic synthesis.

## Study characteristics

Of the 21 studies included in the synthesis, six explicitly explored experiences of managing diabetes alongside SMI and one researched the management of hypertension. The remaining studies focused on general self-management behaviours in people with SMI, with most studies investigating a number of different self-management behaviours (see Table 2).

Of the included studies, four included people with a diagnosis of schizophrenia or schizoaffective disorder [37, 48–50], three included adults with bipolar disorder [35, 41, 54], three included adults with schizophrenia, schizoaffective disorder and bipolar disorder [39, 56, 58], two included adults with any psychotic disorder [55, 57], and one study included young people (aged 18–35) with first episode psychosis [47]. The other eight studies used the term severe mental illness to describe the participants without specifying diagnoses [36, 40, 42–46, 52, 53].

**Table 3. Overview of themes and sub-themes.**

| Theme | Sub-theme |
|---|---|
| The high burden of living with SMI acts as a barrier to self-management | SMI symptoms |
| | Getting out of the house |
| | Side effects of SMI medication |
| | Mental health is prioritised over physical health |
| | Stigma of mental illness |
| Living with co-morbidities presents additional difficulties to self-management | Physical health conditions limited people's ability to engage in physical activity |
| | Taking medication for different things |
| | Interactions between mental and physical health conditions |
| Beliefs, knowledge and attitudes relevant to health conditions and treatment influence self-management | Not knowing what to do |
| | Perceived benefits and consequences of self-management |
| | Beliefs about their capabilities |
| | Attitudes towards self-management |
| | Not accepting their diagnosis |
| Support from others facilitates self-management | Encouragement for self-management |
| | Financial and practical support |
| | Shared experiences |
| | Healthcare staff who care |
| | Lack of support for self-management |
| | Support that was unhelpful |
| Social and environmental factors influence self-management | Living situations and local resources |
| | The company you keep |
| | Self-management is expensive and resource intensive |
| | Emotional effect of the environment |
| Routine, Structure and Planning can promote both positive and negative health behaviours | Forgetting |
| | Habit formation |
| | Having a daily routine and structure |

Most studies were from North America (n = 10) or western Europe (n = 6), with five conducted elsewhere (Australia n = 3, Israel n = 1, Japan n = 1). Reporting of participant demographics was inconsistent across studies. Where reported, the mean age of participants was commonly in the late 40s or 50s.

Participants in the studies lived either in private homes or long-term residential settings, and were mainly recruited from hospital- and community-based healthcare services. Two studies included healthcare professionals as participants [39, 49], while one study included exercise practitioners [52], in addition to people with SMI. The majority of studies collected data through individual interviews, two collected data through focus groups [36, 41] and one analysed data from a panel discussion during a workshop [49]. A variety of approaches to analysis were used, most commonly thematic analysis [36, 44, 46, 53, 55, 58], grounded theory [37, 47, 51, 56] and content analysis [41, 49, 54, 57].

## Review findings

We identified six analytical themes incorporating 26 sub-themes. These are described below, with *'sub-themes'* presented in bold italics. Participant quotations from included studies are provided to illustrate key points and explanatory detail. An overview of the themes and sub-themes can be found in Table 3.

**1. The high burden of living with SMI acts as a barrier to self-management.**   This theme related to the complexity of living with SMI and includes sub-themes that expanded on how the prominence of SMI symptoms can impact a person's motivation and comfort leaving their house, the need to cope with the side-effects of medication, prioritising mental over physical health and the stigma of mental illness.

**'SMI symptoms'** were commonly reported to impact on people's motivation and capacity for self-management and self-care generally [35–37, 40, 42, 45, 46, 48, 50, 51, 53, 56, 57, 59]. This finding is exemplified by this account from a participant from a study about strategies for undertaking physical activity:

*"much of it is that the illness, the paranoia. . .those who are pursuing me, want me to stay at home with my mother and not be out running somewhere else."* [50].

Symptoms of depression in particular were linked to having no energy or motivation to engage in self-management, and a lack of motivation was a commonly reported barrier for people with and without comorbidities to engaging in healthy lifestyle behaviours such as healthy eating and exercise [40, 45, 48–51, 56–58].

The articles also described how poor mental health can become overwhelming and limit peoples ability to engage in even the most basic self-care behaviours:

*"When you are depressed you are just sitting and staring without being washed or anything. It goes beyond everything"* [42].

SMI symptoms and associated problems such as agoraphobia, anhedonia and social anxiety were also identified as barriers to **'getting out of the house'** [45, 48, 58] which itself was reported to impact on self-management behaviours such as physical activity and attending appointments:

*"sometimes it just takes me a long time to get out of the house. I'll just watch TV or sleep."* [43].

Conversely, getting out of the house was viewed as a facilitator of self-management, providing distraction from negative thoughts and improving mood and motivation:

*"it's a positive cycle, so, once you get out you'll just keep going out and out. It becomes addictive. . .it stops you from over-thinking"* [58].

The **'side effects of SMI medications'** were reported in numerous studies to affect people's motivation to engage in self-management behaviours [35, 39, 42, 45, 48–51, 53]. The most commonly reported side effects were lethargy [43, 48, 50, 51, 59] and weight gain, [42, 43, 49, 51, 53] which participants from many studies specifically identified as key barriers to engaging in physical activity. As a participant in one study explained,

*"I think my medications have quite a bit to do with it. . .because if I weren't overweight. . .if I weren't way sedated with medications that I take, I'm sure that I'd be quite active."* [59].

For participants in some studies, trying to lose weight when the medication they took increased their weight seemed pointless,

*"once you are 200 pounds and you can blame it on your medication, how much effort do you really want to put into losing weight or changing your diet. . .?"* [53].

Although other side effects were reported, such as excessive sweating and tremors [41, 48, 51], there was less evidence about how these impacted on self-management behaviours, although in one study participants expressed feelings of not being in control of their body [49].

Several studies found that self-management of physical health conditions and engagement in healthy lifestyle behaviours were neglected as **'mental health is prioritised over physical health'** [36, 37, 46, 53, 59], with some describing how the burden of SMI could make it difficult to focus on other health problems:

*"the first thing you have to do is take care of your schizophrenia, and then take care of your diabetes, because taking care of your diabetes is not going to make you well mentally."* [37].

The **'stigma of mental illness'** was found to impede self-management of SMI and LTCs too [35, 39, 47, 48, 58]. For example, some studies reported how participants expressed concerns that they might be viewed negatively by the public because of their illness, which could impact on their feelings about going out [48, 58], which in one study affected access to diabetes care [39]. Others expressed fears about being classified as having a mental illness, and as a result not wanting to take their psychiatric medication:

*"it took a long time for me to take the medicine because I didn't want to be classified as having a mental illness because I thought I'd be ostracized."* [46].

**2. Living with co-morbidities presents additional difficulties to self-management.** The second theme highlighted how living with multiple different diagnoses, including both mental and physical LTCs, can result in added complexity and difficulties engaging in self-management behaviours. The sub-themes explored how physical health conditions can limit people's ability to engage in physical activity, the difficulties associated with taking multiple medications for multiple conditions, and how physical and mental health can interact and compound barriers to self-management.

While only 7 studies explicitly focused on the experience of living with SMI and co-morbid LTCs, other studies included participants with co-morbidities, although this was not their main focus (Table 2). Therefore 14 studies contributed to this theme [35–37, 39–42, 47, 50–52, 55, 57, 59]. *'Physical health conditions limited people's ability to engage in physical activity'*, this resulted from symptoms such as chronic pain, difficulty breathing and fatigue [35, 52]. For example a participant from one study, who experienced chest pain caused by radiotherapy, stated that

"*when I get the chronic pain I have on my left side, I can't move or walk.*" [35].

A participant from another study explained that

"*I have bad arthritis and that prevents me from being able to do most exercise.*" [52].

Other studies described how being overweight (which for many was linked to SMI medication–see Theme 1) caused breathing difficulties, and together these limited their ability to engage in physical activity [44, 59].

*'Taking medications for multiple things'* was also reported as a barrier to self-management [35, 37, 41, 46]. In two studies, taking "*too many pills*" [35, 41] made it hard for participants to keep track of the various medications they were taking or to work out which medication may be causing side effects, thereby impacting on their medication adherence:

"*the side effect was personal and I didn't know what pill might be doing it, and so I'd stop one medication at a time to see which one it was*" [41].

How medications interacted with each other, and with mental illness, was a concern for participants in some studies [46, 55]

"[I] *take insulin and that interacts with bipolar and causes mood swings too.*" [46].

Additionally, in one study, taking certain medications was reported to prevent participants from being able to use particular smoking cessation medications [55].

*'Interactions between mental and physical health conditions'* influences self-management too [35, 36, 39, 41, 42, 46, 50, 51, 55, 57, 59]. For example, one study that included participants with diabetes described experiencing fatigue from increasing blood sugar levels, which in turn had a negative impact on their mental health (which affected motivation for self-management —see Theme 1).

"*when your blood sugar is 300 or 400, you get tired and groggy. . .it does have an effect on my mental condition*" [37].

Poor mental health was reported to exacerbate diabetes as well:

"*When I'm anxious, my blood sugar gets very high and difficult to manage. And when my bulimia is bad and I vomit, my blood sugar is also fairly difficult to manage. Thus, it can be extremely complicated to make everything stick together.*" [42].

Across these studies, participants mainly described the negative impact of mental health on LTC self-management [35, 36, 46, 50, 51, 57, 59]. However, some studies reported that

participants were able to draw on their experience of managing their mental health to manage any new conditions they developed,

> "*I've been stable mentally for 15 or 20 years. . .so I had a good jump on the diabetes when it started happening. I could take the medicine and remember to take it, and watch my sugar, and it would be ok.*" [37].

Similarly, others reported that improvements in physical health as a result of managing diabetes impacted positively on their mental health as well [37].

**3. Beliefs, knowledge and attitudes relevant to health conditions and treatments influence self-management.**   The third theme related to how beliefs, knowledge and attitudes towards SMI and other LTCs, influenced engagement in self-management. The sub-themes related to how a lack of knowledge can make it difficult for people to know how to engage effectively in self-management behaviours, how perceived benefits and consequences, belief in capabilities, and attitudes can influence self-management, and finally the need for people to accept their diagnosis in order to change their behaviour.

Participants in several studies talked about '***not knowing what to do'***, although this varied significantly within and across studies, and by self-management behaviour [35, 36, 39, 40, 46, 47, 50–53, 57]. While studies reported participants had a general lack of knowledge about self-management behaviours [47, 52] some of this uncertainty was due to contradictory advice or misinformation by family and friends [35, 46]:

> "my *family is always telling me 'I don't think you need to take the medication. . .and people telling me 'girl you don't need that medicine, just all you need to do is cut the stress in your life, you don't need the medicine. . . and that makes me say 'okay, I don't need it no more.'*" [46].

Although participants in the included studies seemed to have some awareness of healthy eating recommendations [40, 53], many struggled to comply because of gaps in their knowledge [40, 53]. Others struggled with understanding and interpreting food labels even when these were explained to them [35, 51]. One study reported that participants' understanding of how to manage their metabolism caused them to have strange eating habits like '*eating bread and lemon water*' [57]. Participants in other studies were unsure about the level of physical activity considered sufficient for managing their conditions:

> "*I understand the little I do actually has no effect. Such short sessions are pointless, so little. They should be long sessions. . .if you swim, it ought to be a kilometre.*" [50].

Some studies described how not understanding the cause of their bipolar disorder meant participants struggled to effectively manage their symptoms [46, 57]:

> "*I still don't understand what constitutes it. To understand it is the first issue and since I don't understand what symptoms are, I gotta first know 'em before I can say I'm aware of 'em.*" [46].

While one study reported how participants with psychosis changed harmful health behaviours like "*cutting down on snuff*" after increasing their knowledge by "*reading mindfulness books, looking up on the web and getting several tips.*" [57]. Another study highlighted some participants felt they lacked information on the health consequences of their condition and anti-psychotic medication:

"'*I didn't know I needed to change my lifestyle, no one told me. . . I didn't [have to] worry about not being healthy and not feeling in shape before taking the [antipsychotic] medicine*'" [47].

Beliefs about the '***perceived benefits and consequences of self-management***' were reported to influence several self-management behaviours including taking medications [35, 37, 40, 41, 46, 49, 50, 56], physical activity [40, 43, 47, 52, 57, 58], smoking cessation [54–57], and healthy eating [36, 53, 56]. For example, several studies reported that participants took their psychiatric medication despite negative side effects such as increased appetite and weight gain because they believed it treated their mental health symptoms [37, 41, 46, 49, 50, 57]. The decision to take medication and which medication to take was sometimes arbitrary:

"*sometimes I'm going to take my psych meds today and sometimes I don't want no psych meds today, but I feel it's really important for me to take my blood pressure medicine. . .so it's not all or none, sometimes you take one and not the other. It could go either way.*" [41].

Four studies that focused on physical activity highlighted how participants believed that being physically active had many benefits including increased feelings of happiness, 'freedom and independence', extra stamina, improving mental illness symptoms, weight loss [47, 50, 57, 58], and an opportunity to get out of the house [58]. However, some of these studies also reported fear of physical injury and the feeling of 'not getting much out of it' caused participants to be hesitant of beginning physical activity [50, 58].

Stopping smoking was perceived to bring both benefits and potential drawbacks, with smoking described as a source of pleasure and comfort, having a calming effect and improving mood, reducing irritability and fear, and stabilising weight [54–56]. A participant in one study expressed,

"*it calms you down . . .well I am worried that I might feel unwell if I stop-that is another reason that I dinnae want to stop.*" [56].

However, for some, the financial cost of smoking was a facilitator to quitting [54, 57] and improved confidence, mood, sleep, appetite and overall self-esteem were noted as benefits of quitting [54].

The literature described how an increased awareness of the risk of diabetes complications caused some participants to change their eating habits [53]. Others, despite being aware of their unhealthy diet believed that they were unlikely to develop health problems [36, 53, 56]. In two studies including participants with SMI and diabetes [37, 53], seeing family members experience diabetes and diabetic complications encouraged participants to manage their own condition better:

"*my mom has diabetes worse than me. . .just seeing what she has to do every day, which is take shots, it's just something I don't want to do, I don't want to be on insulin.*" [37].

The included articles demonstrated that participants' '***beliefs about their capabilities***' influenced their self-management, particularly physical activity, healthy eating and smoking cessation [40, 48, 50–53, 56–58]. Some studies reported that participants had low faith in their abilities to engage in physical activity [48, 50–52, 57, 58]. For example, a participant in one study expressed his lack of confidence to swim as well as he used to before the onset of his mental illness:

"*when I go to the pool, I do one length and I'm [like] 'oh, I can't do this'..*" [48].

In relation to healthy eating, participants expressed difficulty with controlling their sugar intake [36, 40], and adhering to dietary and healthy eating recommendations [42, 53, 56, 57]:

"*I get food cravings and can't just pull myself together to eat healthy food*" [42].

Eight studies reported that '***attitudes towards self-management*** influenced behaviour' [36, 37, 50, 53, 56–58]. Although managing their conditions was sometimes tough, participants in these studies had 'dreams and hopes for the future', felt a sense of responsibility and desired to 'do the best they can' to manage their physical and mental health conditions [37, 57]. Participants from some studies used positive attitudes and words as 'power tools' to maintain physical activity and healthy eating when it became unenjoyable:

"*It is helpful to be positive and self-confident, telling yourself that it has worked before and will work again.*" [50].

Despite the positive attitudes, there were negative attitudes towards healthy eating [53, 56], physical activity and monitoring of symptoms [36].

Three studies reported that self-management was impeded for some participants because of '***not accepting their diagnosis***' [36, 40, 47]. Some participants in these studies initially experienced difficulty in accepting their diabetes diagnosis, which invariably had a negative impact on their ability to manage and seek help for their conditions [36, 40]:

"*I went to a diabetes education program and at the time I didn't learn much because I was in huge denial.*" [36].

For participants in some studies, an obstacle to engaging in physical activity [50, 57], eating healthily [53, 56, 57] and smoking cessation [54–57] was a 'lack of readiness / urgency to change'. These studies described how participants often made plans for a future healthy lifestyle but never found the right time [57].

**4. Support from others facilitates self-management.** The fourth theme explored how support from others could facilitate engagement in self-management for people living with SMI. Three sub-themes highlighted how support can positively influence self-management, including providing encouragement, financial and practical support, and connecting with others through shared experience. However, there were also two sub-themes that related to barriers around support from others, namely the lack of support people with SMI receive for their self-management, and unhelpful support that made self-management more difficult.

Many studies identified support from others as a facilitator of self-management [40–43, 48, 50, 54, 57, 58]. Support came from various sources and in different guises, for example as '***encouragement for self-management***' from family members, friends or healthcare staff for taking medication [41, 42], eating healthily [40], engaging in physical activity [48, 50, 57, 58], smoking cessation [54] or losing weight [43]. Staff at local community mental health centres were identified as particularly important sources of encouragement in several studies [40, 43, 48, 50].

Family members, home-care staff, and community centres were also described as sources of '***financial and practical support***' [37, 39, 40, 42, 52, 57]. For example, several studies [38, 40, 41, 50, 52] highlighted how participants were supported to maintain routines and provide structure, especially in terms of eating habits when mentally unwell:

*"I wouldn't eat right. . .I can't because of the voices. I would go without eating for 2 or 3 days because of the voices. I moved in with my brother and he took care of me. I couldn't do it without him."* [37].

In relation to physical activity, a participant in one study stated that:

*"it helps if someone else suggests something, goes with you. If you are depressed and don't take these initiatives, but have someone who gives you a push, then it [exercise] suddenly feels like fun."* [50].

Participants in four studies described the importance of receiving support from others who had **'shared experiences'** of managing similar conditions, which they identified as facilitating better self-management [36, 48, 50, 52]. For example, =

*"my brother-in-law, he has diabetes, so if I need anything, I'll call him and ask him.".* [36].

While a participant in another study reported that:

*"I can get on with a group of people who have mental health problems because I understand what they are going through."* [48].

There were mixed reports about the role of healthcare staff in providing support for self-management [35, 36, 39, 40, 42, 43, 46–49, 51, 53, 56–58], although having **'healthcare staff who care'** was commonly reported across the studies to make the difference between feeling supported or not. Some participants also found it beneficial when staff took interest in both their physical and mental health [43, 57], and in one diabetes study, participants identified healthcare staff as a major source of diabetes education and help, especially with taking medication and monitoring symptoms [40]. In other studies, participants felt healthcare staff needed to explain medical results better and avoid medical jargon [45, 59–61]. It was also believed that poor communication from staff [35, 36, 39, 42, 46, 47, 49, 51], and a lack of continuity of care, were barriers to accessing support for both mental and physical health,

*"Dr B reads Dr A's notes from endocrinology and that's the only part he reads in my chart. I don't believe he goes anywhere else regarding my mental health. . .the doctors need to grasp the whole picture."* [35].

Despite the value of support from others, numerous studies reported a **'lack of support for self-management'** [35–37, 39, 40, 46, 50, 51, 57], with some participants in these studies reporting that they felt isolated and lonely and had no family or friends to support them [35, 37, 39, 40, 46, 50, 51]. This was sometimes found to be a barrier to healthy eating and engaging in physical activity [51]. This lack of support was also described in studies with participants who felt that their mental illness was not well understood by others or that family and friends avoided them [46] or had given up on them [54].

Several studies also described **'support that was unhelpful'**, although negative accounts were less commonly reported. For example, participants in some studies found it difficult to eat healthily if family members didn't:

*"my husband teases me sometimes cause he likes his sugar and I'm sitting there and he's eating it."* [36].

The literature highlighted that some participants received contradictory messages from healthcare staff that prevented them from engaging in self-management, for example receiving confusing information about healthy eating [39, 51], and mixed messages about whether they needed to make changes to their health [56].

**5. Social and environmental factors influence self-management.**   The fifth theme identified the influence of social and environmental factors on self-management behaviours in adults with SMI. The sub-themes highlighted the importance of living situations and local resources in people's ability to manage their health, the role of peer groups and social circles in engagement in health behaviours, that engaging in self-management for LTCs can be resource intensive and expensive, and that a person's environment can have an emotional effect that in turn influences self-management behaviours.

Several studies expanded on how the social circle and environment of participants could affect their behaviour. Participants in these studies described how their ***living situations and local resources*** influenced their ability to manage their health [35, 36, 39, 40, 46, 50–53, 55, 58, 59]. Some struggled to find a place to live, contributing to depression and other mental illness symptoms [46].

'Exposure to unhealthy food' was reported to hinder efforts to eat healthily [35, 36, 39, 53]. This was described as a result of social situations, having easy access to unhealthy foods, or limited food choices, for example in group homes or residential facilities,

> "*If you go to McDonald's and they've got the dollar drinks on, you're going to drink it. I tried diet Coke, didn't like it, poured it out and put the regular stuff in. That's what drives my sugars up.*" [36].

In one study participants reported how they relied on food banks and community kitchens, and this limited their food choices [39]. Another study of people with SMI in a group home also reported a lack of facilities to prepare healthy meals:

> "*where I lived at didn't have an oven. . .so I'd just eat cold stuff out of the refrigerator, out of the cabinet, and go out. . .eat mostly cheap stuff.*" [53].

There was also a culture of 'not doing a lot' in the facilities and food was served as a distraction to help cope with the lack of physical activity [51].

The external environment of participants in these studies, such as weather conditions, distance to a gym, lack of sports equipment or neighbourhood safety, were identified as barriers to engaging in physical activity [50–52, 59]. Some participants also talked about their emotions associated with these factors, which appeared to influence their behaviours:

> "*I fell last winter on my way to the garbage disposal and hit my hip and shoulder really hard, I was terrified. . .it can really make you afraid when you have fallen like that and then it is easy to avoid going out.*" [50].

Participants in several studies described that ***'the company you keep'*** can have a negative influence on self-management, in particular efforts to stop smoking or drinking alcohol [36, 54, 55]. Studies described social impediments to quitting such as being in the physical presence of other smokers [54, 55], with a participant stating that others would say to him "dude if you need to smoke, smoke." [54]. Participants in another study further reported that cultural expectations made it difficult to eat healthily:

"*Eating is addicting in Latino culture and when I visit my sister or my mom, they always give me food. . .and you feel the pressure to eat*" [53].

Several studies illustrated that '**self-management is expensive and resource intensive**'*,* particularly for physical health conditions, including diabetes [38, 40, 46, 50–53, 56, 57, 59]. Many participants in these studies described how they could not afford transport to get to appointments [35, 36, 38, 42, 46], testing strips to monitor their blood sugar levels [35, 36, 38, 40, 46], healthy food [35, 38, 39, 51, 53, 56], medication [38], and gym membership, clothes and equipment [50–52]. Some described how self-management behaviours were less of a financial priority:

"*Gyms are too expensive and I'm a single mum . . . paying for rent as well . . .*" [52].

Another factor that influenced self-management was the '**emotional effect of the environment**'**.** The included studies highlighted how social anxiety [42, 43, 48, 50, 52, 57, 58], boredom [51, 56] and emotional attachments [55] hindered self-management. Participants in five studies reported how interacting with others made them anxious and that in turn posed a barrier to being physically active. Many who experienced agoraphobia found it difficult to get out of the house:

"*I guess I kind of struggle with being outside a lot. I feel safer inside.*" [43].

The fear of being perceived negatively by others prevented people from participating in physical activities [48, 50, 57, 58], particularly in relation to communal settings such as gyms:

"*You go to a gym . . . most of them are young people and they're looking at you sideways thinking this is a big lady that's come in. . . they've all got their skinny tight little bums and they look at you like "ergh what are you doing here?", I'd rather stay away.*" [52].

Participants in two studies reported that being bored made them eat a lot and unhealthily: "*I tend to eat a lot since I'm bored in this facility.*" [51]. In one study, a participant explained how an emotional attachment to smoking posed a barrier to their smoking cessation efforts:

"*maybe you had some memory that had to do with cigarettes like you share the cigarettes with a person you love or something. . .or maybe there's just some weird emotional attachment to it.*" [55].

**6. Routine, structure and planning can promote both positive and negative health behaviours.**   The final theme explored how routine, structure and planning could facilitate not only self-management behaviours, but also behaviours that can harm health, such as smoking. The sub-themes looked at the role of forgetting as a barrier to adhering to medication regimens, the way in which habit formation can promote self-management behaviours but can also make harmful behaviours more difficult to change, and also how having a routine and daily structure could facilitate engagement in self-management.

**'Forgetting'** was a commonly reported barrier to adhering to medication regimens [36, 37, 40, 41, 54]. Forgetting to take medication for some participants was associated with the competing demands of life such as their jobs [41] the complexity of managing multiple morbidities and medications [36, 41], and medication schedules that were inconvenient [36]:

"*If I forget to take the bipolar meds then I forget to take the one for my blood pressure. If I don't take one then I'm not going to take the other one.*" [41].

Other studies found that when participants were mentally unwell, they were more likely to forget to take their medication or repeat doses [40], or to check blood glucose levels and eat regularly [40]. Despite this, some reported never forgetting to take their medications [40]. One study described how using nicotine replacement to quit smoking was challenging, with a participant describing how they had forgotten their nicotine patches:

"*it was just a difficult time and without the patches, I've almost felt panicked, in a way.*" [54].

Most of the studies describing '***habit formation'*** were about smoking [54–56]. Participants in these studies found it difficult to quit because smoking had become habitual, with one participant describing it as 'an extension of me' [54] and another stating

"*when I wake up, I smoke, after I eat, I smoke. . .when I drive, I smoke. . .waiting on the bus, I wanna smoke.*" [55].

Participants in another study also reported that their eating habits conflicted with eating healthily:

"*I'm supposed to eat lots of little meals instead of big meals. That's hard to do for me. Just so different from the way I've always ate all my life. . ..*" [53].

'***Having a daily routine and structure'*** was identified in numerous studies as an important strategy used by participants to embed self-management behaviours into their lives [40, 43, 46, 48, 50, 53–55, 57, 58]. Conversely, other participants in these studies who lacked structure or routine struggled to undertake self-management behaviours, especially healthy eating and smoking cessation [53, 55]. For some it was difficult to maintain a healthy diet because eating healthily was viewed as less convenient:

"*I grab whatever is around. . .go for the easy, the junk, because I don't feel like actually having the time to sit down and prepare, and make, and clean up and because then it is not even just that simple*" [53].

While another study noted how getting tired of routines had an impact on medication taking:

"*a lot of times I stop taking my medication because I get tired of just the routine of taking medication. I'll just get up one morning and just say 'I ain't taking it'. . .then a few days go by. . .*" [46].

## Discussion

This systematic review and qualitative synthesis has shown that self-management of physical health in the context of SMI is determined by a complex mix of factors associated with the impact of mental and physical health. These factors include the symptoms of, and medications for, SMI, beliefs about the merits of self-management and how best to perform self-management tasks, and the environmental and social day-to-day experiences of people with SMI.

The burden of living with SMI encapsulates both the symptom burden and treatment burden of the condition, but also the social consequences of mental illness. Participants across studies consistently reported that the management of SMI was prioritised over self-management for other health concerns. This supports evidence that people living with complex

healthcare needs prioritise the self-management of a dominant condition, particularly if it is a condition that is not fully controlled or can cause significant disruption to daily life, such as SMI [60].

The evidence reviewed in this synthesis suggests that barriers to engaging in self-management behaviours for people with SMI include a lack of belief in their capability and knowledge to engage in self-management, the stigma of mental illness, and an inability to accept their diagnosis. Recognition, and understanding the significance of, diagnoses of physical health problems has been shown to underpin people's willingness to engage in self-management behaviour in non-SMI populations. Furthermore, the relationships between these issues are complex and bidirectional, with evidence showing that poor health outcomes and paternalistic decision-making are associated with low self-efficacy, high self-stigma, and low levels of education [61].

While the experiences of living with SMI influenced engagement with self-management, resources and support also shaped participants' experiences. Participants reported that a lack of financial and environmental resources limited their ability to engage in behaviours such as eating healthily and engaging in physical activity. People with SMI experience higher levels of socioeconomic deprivation and are more likely to live below the poverty line [62], suggesting these barriers to self-management are pervasive in this population. The evidence also highlighted the importance of support from others, socially, professionally or practically. People with SMI experience high rates of social isolation and loneliness [63], a problem that is not only closely linked to the symptoms of SMI and the associated stigma, but also socioeconomic deprivation [64].

Our findings emphasise the importance of effective communication between healthcare providers and patients. Poor communication was described as hindering self-management and resulted in confusion and uncertainty. Furthermore, ineffective communication has been identified as a contributing factor to diagnostic overshadowing in clinical settings, where healthcare professionals attribute symptoms resulting from physical illness to SMI [10], leading to inadequate assessment of physical health issues. Participants also reported that some relationships with healthcare professionals were passive, with limited evidence of shared decision-making. While shared decision-making is recommended in mental health settings, this does not always translate into practice, as patients with schizophrenia frequently report that they do not feel involved in their treatment decisions [65]. People with SMI commonly report being excluded from decision making, particularly in relation to psychiatric medication [66]. Whilst the literature suggests healthcare professionals hold positive attitudes towards supportive self-management to improve patient outcomes, this does not always translate into intention and practice. Healthcare professionals also demonstrate uncertainty around what self-management is and how it can best be supported, therefore more education is needed for staff as well, to ensure adequate understanding of self-management and the healthcare professionals role in providing support [67].

## Strengths and limitations

Qualitative evidence syntheses have been described as a useful 'technology' for bridging the gap between evidence and decision-making [34]. By synthesising data from available qualitative studies, this synthesis offers an in-depth and comprehensive overview of the lived experience of self-management for people with SMI [68], and provides novel insights and understanding about factors influencing self-management in people with SMI with and without long-term conditions. In this sense our work methodologically maps to exploratory and modelling approaches favoured by the MRC Framework and the Science of Behaviour Change programme to inform the design of complex interventions [69, 70].

We did not use conventional quality assessment tools (e.g. CASP) to scrutinise methodological quality of included studies. The value of quality assessments using narrow definitions about methodological quality in qualitative reviews is debatable [71, 72]. There is now increasing recognition that relevance rather than quality alone is a critical factor that underpins decision making about the merits and utility of data in qualitative evidence synthesis [73]. In this sense our interpretative judgments about the utility and relevance of included studies to contribute to the synthesis was based on data richness and not quality based on scoring systems that often discount attributes related to richer or 'thicker' data. This is an approach we have successfully deployed in previous qualitative evidence synthesis whereby the use of thicker or richer data approximates critical appraisal [74].

As with all systematic reviews, there is a risk that potentially eligible studies have been missed. However, our search strategy was comprehensive and inclusive, and study selection methods were designed to reduce the risk of reviewer bias and error, making it less likely that our results would be substantially changed by inadvertently missing studies.

We excluded papers published in languages other than English as we did not want to risk losing the meaning of the participants' quotes by having to retrospectively translate the individual primary studies. It is important to bear this in mind when considering the extent to which our findings can be generalised to other countries or settings.

Similarly, care needs to be taken in applying our findings to different LTCs. This review identified only a limited number of studies on people with SMI and a physical LTC. Most of these studies focused on people with SMI and diabetes, highlighting a dearth of qualitative research exploring the experiences of SMI and other long-term conditions.

## Implications for practice

Policy and practice that aims to support people with SMI manage co-morbid physical LTCs should be done in the context of the unique difficulties people with SMI experience as a result of their mental illness. Developing person-centred education and support programmes, tailored to the needs of this population, could help promote self-management of both SMI and co-morbid LTCs. This evidence synthesis highlights that support programmes should account for the additional burden of SMI, including symptoms such as anxiety and poor motivation, difficulties people experience leaving their house, and the stigma of mental illness.

It is crucial that shared decision-making is promoted and used to support management of co-morbid LTCs in people with SMI. People living with SMI should be supported to actively participate in managing their health and making decisions about their treatment. In order to facilitate this healthcare professionals need to receive education and training about supported self-management, taking into account the unique issues experienced by people living with SMI and co-morbid LTCs.

Finally, financial resources and access to social support were highlighted as important facilitators of self-management. This evidence synthesis highlighted how access to resources and positive social support facilitate self-management for people with SMI. However, it is important that any policy that aims to support people with SMI manage their health accommodates, acknowledges and seeks to address limits that stem from socioeconomic deprivation.

## Implications for research

We found a wealth of relevant evidence which largely provided rich data on the experiences of self-management for people with SMI, however limited evidence on the self-management of co-morbid physical LTCs with the exception of a few studies on type 2 diabetes. Future

research should aim to address this gap by exploring the experiences of people with SMI and other co-morbid LTCs, other than type 2 diabetes. Additionally, demographic characteristics of participants in the included studies were poorly reported, and there is a need for more complete and consistent reporting of participant characteristics, including important demographic factors such as ethnicity and gender. This will facilitate a better understanding of how intersectionality, how the relationship between multiple demographic categories that result in systemic discrimination, underpins many of the health inequalities faced by people with SMI.

Furthermore, research is needed to identify what barriers, facilitators and strategies can be targeted or adopted in a complex intervention, that can improve self-management and ultimately improve outcomes, such as morbidity and mortality, in people with SMI and LTCs. Any work developing interventions to support self-management in people with SMI should be interdisciplinary and include individuals with lived experience. This is important to ensure new interventions and programmes address the specific challenges faced by this population, especially around effective communication and linked-up healthcare provision.

## Conclusion

Living with SMI not only directly influenced people's experiences of self-management due to the associated symptoms and treatment, but also indirectly through the ways in which SMI affected other areas of the person's life. A number of facilitators for self-management were identified through the synthesis, including having a routine, having adequate social support and encouragement for self-management, engaging in shared decision making, and having access to resources necessary to engage in self-management behaviours. However, the experience of living with SMI acted as a barrier to self-management, and compounded other existing barriers associated with LTCs. As people living with SMI are more likely to experience paternalistic healthcare, diagnostic overshadowing, stigma, socioeconomic deprivation, and social isolation, they face additional barriers to self-management compared with people who do not have SMI. These barriers are closely interrelated and mirror the complex relationship between mental and physical health.

## Supporting information

**S1 Checklist. Prisma 2020 checklist.**
(DOCX)

**S1 Appendix Revisions to systematic review protocol.**
(DOCX)

**S2 Appendix. Ovid Medline search strategy.**
(DOCX)

**S3 Appendix. Data richness scale used for quality appraisal (Ames et al., 2017) [32].**
(DOCX)

**S4 Appendix. Summary of studies eligible for review.**
(DOCX)

## Acknowledgments

We would like to acknowledge the work of the wider DIAMONDS research team, which includes (in addition to the authors) Simon Gilbody, University of York (MHARG), Co-investigator, Patrick Doherty, University of York (HoD, DoHS) Co-investigator, Nicky Traynor,

University of York (MHARG), Programme Administrator, Sarah McCardle, (MHARG), Research Administrator, Jude Watson, University of York (YTU), Co-investigator, Tim Doran, University of York, Co-investigator, Simon Walker, University of York (CHE), Steve Parrott, University of York, Catherine Hewitt, University of York (YTU), Co-investigator, Michael Crooks, Hull York Medical School, Sally Carling, PPI Member, Keith Double, PPI Member, Angela Ross, PPI Lead, David Osborn, University College London, Kelly Barker, Bradford District Care NHS Foundation, John Hiley, Bradford District Care NHS Foundation, and Angela Moulson, Bradford District Care NHS Foundation.

## Author Contributions

**Conceptualization:** Peter Coventry, Ramzi Ajjan, Sarah Alderson, Sue Bellass, Jan R. Boehnke, Richard Holt, Rowena Jacobs, Ian Kellar, Charlotte Kitchen, Jennie Lister, Emily Peckham, David Shiers, Najma Siddiqi, Judy Wright, Jo Taylor.

**Data curation:** Claire Carswell, Jennifer V. E. Brown, Judy Wright.

**Formal analysis:** Abisola Balogun-Katung, Claire Carswell, Charlotte Kitchen, Jo Taylor.

**Investigation:** Abisola Balogun-Katung, Jennifer V. E. Brown, Peter Coventry, Charlotte Kitchen, Jo Taylor.

**Methodology:** Abisola Balogun-Katung, Claire Carswell, Jennifer V. E. Brown, Peter Coventry, Najma Siddiqi, Judy Wright, Jo Taylor.

**Project administration:** Jennifer V. E. Brown.

**Resources:** Najma Siddiqi.

**Supervision:** Najma Siddiqi.

**Writing – original draft:** Abisola Balogun-Katung, Charlotte Kitchen, Jo Taylor.

**Writing – review & editing:** Abisola Balogun-Katung, Claire Carswell, Jennifer V. E. Brown, Peter Coventry, Ramzi Ajjan, Sarah Alderson, Sue Bellass, Jan R. Boehnke, Richard Holt, Rowena Jacobs, Ian Kellar, Charlotte Kitchen, Jennie Lister, Emily Peckham, David Shiers, Najma Siddiqi, Judy Wright, Ben Young, Jo Taylor.

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
