## [Decision Letter · Decision Letter 0]

21 Jul 2021

PONE-D-21-18104

Exploring the facilitators, barriers, and strategies for self-management in adults living with severe mental illness, with and without long-term conditions: a qualitative evidence synthesis

PLOS ONE

Dear Dr. Claire Carswell,

Thank you for submitting your manuscript to PLOS ONE. After careful consideration, we feel that it has merit but does not fully meet PLOS ONE’s publication criteria as it currently stands. Therefore, we invite you to submit a revised version of the manuscript that addresses the points raised during the review process.

We ask your team to address comments raised by the peer reviewers' about the qualitative review's methods and presentation of findings. Please refer to the detailed comments at the end of this section.

We look forward to receiving your revised manuscript.

Kind regards,

Eleanor Ochodo

Academic Editor

PLOS ONE

“This article presents independent research funded by the National Institute for Health Research (NIHR).”

“NS received an NIHR Programme Grant for Applied Research (Grant number: RP-PG-1016-20003) by the National Institute for Health Research:

https://dev.fundingawards.nihr.ac.uk/award/RP-PG-1016-20003

3. One of the noted authors is a group or consortium DIAMONDS research team. In addition to naming the author group, please list the individual authors and affiliations within this group in the acknowledgments section of your manuscript. Please also indicate clearly a lead author for this group along with a contact email address.

Reviewers' comments:

Reviewer's Responses to Questions

**Comments to the Author**

1. Is the manuscript technically sound, and do the data support the conclusions?

Reviewer #1: Yes

Reviewer #2: Partly

2. Has the statistical analysis been performed appropriately and rigorously? 

Reviewer #1: Yes

Reviewer #2: N/A

3. Have the authors made all data underlying the findings in their manuscript fully available?

Reviewer #1: Yes

Reviewer #2: Yes

4. Is the manuscript presented in an intelligible fashion and written in standard English?

Reviewer #1: Yes

Reviewer #2: Yes

5. Review Comments to the Author

Reviewer #1: The research article is presented in an intelligible fashion and written in standard English. The data presented in the paper support the conclusions made and address the hypothesis from the originally registered report protocol. All underlying data is fully available in the manuscript.

There are deviations from the registered protocol, and these deviations have been clearly described in the manuscript and are appropriate for the study. However, there was an additional exploration of other participants (those who provide care or support to people with SMI) not outlined in the registered report protocol. Inclusion of these participants is reasonable but should be acknowledged as part of the revisions to the registered protocol.

The analysis of extracted data is described in sufficient detail and are methodologically sound.

Minor revisions

• Include other participants explored in the study (those who provide care or support to people with SMI) to the “revisions to systematic reviews” table.

• The Author should fill in the missing details in the Methods section, paragraph one, last sentence on Page 5 of the manuscript before publication.

“Findings from quantitative studies have been reported xxx [ to add details nearer submission depending on where we are up to with that paper.”

Reviewer #2: Thank you for the opportunity to review this manuscript!

ABSTRACT:

Background: The evidence gap could be rephrased. Only mentioning barriers makes it seem like this QES is solely about barriers.

Aim: The aim and title seem mismatched. Is the review including information about long-term management of LTCs only or not?

Methods: The word ‘determinants’ is vague and could be replaced with “barriers, facilitators and strategies”. This comment is applicable across the whole manuscript. Why did the authors only define self-management in the abstract and not “severe mental illness” and “long-term conditions”? Additionally, one or two sentences could be added on the actual review processes, e.g. screening, data extraction, analysis, methodological quality assessment, and if these were done in duplicate?

INTRODUCTION:

The authors should clarify what the purpose of this QES is. For example:

“Exploring the facilitators, barriers, and strategies for self-management in adults living with severe mental illness, with and without long-term conditions: a qualitative evidence synthesis” (title)

“We therefore systematically reviewed and synthesised qualitative evidence about the experiences of self-management in people with SMI to understand how these may impact on LTC management in this population.” (rationale)

METHODS:

Deviations from the protocol: It is not unusual for review authors to deviate from the protocol, but it potentially concerning that there were several deviations, as this potentially indicates that the review question and eligibility criteria were not sufficiently clarified before the start of the review.

Authors have left a note within the text. “Findings from quantitative studies have been reported xxx to add details nearer submission depending on where we are up to with that paper].”

Eligibility criteria:

The authors should have used a question framework like SPICE or SPIDER to organise the eligibility criteria, given that the purpose of the review is still unclear.

The authors mention that they will include qualitative studies of any design. Does this include mixed method studies with qualitative data?

It is unconventional to summarise the eligibility criteria in a table. The eligibility criteria are an important part of the review and should be described in detail in the text.

Data extraction:

Are the authors able to share the data extraction form they used or mention the aspects the extracted?

Quality appraisal:

Did the authors use a methodological quality appraisal tool (e.g. CASP or QARI)? What is currently described in this section has to do with data richness. A ‘rich’ study is not by default of ‘good’ quality.

RESULTS:

It is great that the authors have included quotes from the primary studies, but it sometimes reads as though the quotes are from the authors own primary research. This is a higher-level analysis of primary research and should be presented as such.

DISCUSSION:

Strengths and limitations:

“The review provides novel insights and understanding about factors influencing self-management in people with SMI, which can inform critical thinking about how best to design tailored self-management interventions for this population.” This needs to be explained in the context of existing literature.

6. PLOS authors have the option to publish the peer review history of their article (what does this mean?). If published, this will include your full peer review and any attached files.

Reviewer #1: No

Reviewer #2: No

---

## [Author Response · Author response to Decision Letter 0]

5 Aug 2021

Dear Editor and Reviewers

Thank you so much for taking the time to review our submitted manuscript and providing informative and constructive comments that we believe have improved the over all quality of the manuscript. 

In relation to the comments from the editor, we have changed the formatting of the paper to match the template provided by PLOS One. We have also removed any reference of funding from the submitted manuscript, and we would appreciate if the funding statement could be changed to read the following:

This paper reports work undertaken as part of the DIAMONDS programme, which is funded by the National Institute for Health Research under its Programme Grants for Applied Research (project number RP-PG-1016-20003). Peter Coventry is part funded by the UK Research and Innovation Closing the Gap Network+ [ES/S004459/1].

We have also provided a list of names from the larger DIAMONDS Research Group in the acknowledgements section according to your guidance. 

I have provided a list of the reviewers comments and our responses in the below table. 

Response to reviewers comments

We want to thank both of the reviewers for providing such constructive comments, and for taking the time to read and feed back on our manuscript. 

Reviewer 1

Include other participants explored in the study (those who provide care or support to people with SMI) to the “revisions to systematic reviews” table. 

This has been added to the table.

The Author should fill in the missing details in the Methods section, paragraph one, last sentence on Page 5 of the manuscript before publication.

“Findings from quantitative studies have been reported xxx [ to add details nearer submission depending on where we are up to with that paper.” 

This has been updated.

Reviewer 2

The evidence gap could be rephrased. Only mentioning barriers makes it seem like this QES is solely about barriers. 

We have now stated throughout the manuscript that this review focuses on barriers, facilitators and strategies for self-management of long-term conditions and SMI (rather than just barriers alone)

The aim and title seem mismatched. Is the review including information about long-term management of LTCs only or not? 

We have now rephrased the title and the aim to highlight the review focuses on studies that evaluated self-management strategies in people with SMI either with or without a long-term condition. 

The word ‘determinants’ is vague and could be replaced with “barriers, facilitators and strategies”. This comment is applicable across the whole manuscript. Why did the authors only define self-management in the abstract and not “severe mental illness” and “long-term conditions”? 

Additionally, one or two sentences could be added on the actual review processes, e.g. screening, data extraction, analysis, methodological quality assessment, and if these were done in duplicate?

We agree that the term ‘determinants’ doesn’t capture the full range of factors that might underpin self-management behaviours and we have where appropriate used the phrase ‘barriers, facilitators and strategies’ to better define the focus of the analysis. 

We also agree that defining only one core concept in the abstract is inconsistent, and the definition for SMI and LTC were not provided as there is a limit of 300 words for the abstract. Due to the word limit and the request for more information on other aspects of the methods in the abstract, the definition for self-management has been removed to make the abstract more consistent. 

We have now included an additional sentence about screening, and the abstract mentions the purposive sampling according to data richness and the synthesis. 

The authors should clarify what the purpose of this QES is. For example:

“Exploring the facilitators, barriers, and strategies for self-management in adults living with severe mental illness, with and without long-term conditions: a qualitative evidence synthesis” (title)

“We therefore systematically reviewed and synthesised qualitative evidence about the experiences of self-management in people with SMI to understand how these may impact on LTC management in this population.” (rationale)

This has been clarified in the rationale so it is more consistent with the title and the purpose is defined as per the reviewer’s useful suggestion. 

The authors should have used a question framework like SPICE or SPIDER to organise the eligibility criteria, given that the purpose of the review is still unclear.

The authors mention that they will include qualitative studies of any design. Does this include mixed method studies with qualitative data?

We are not convinced that using SPICE or SPIDER post-hoc would add value at this stage. We had previously outlined the eligibility criteria for the review in Table 1. We appreciate that not all items were clearly defined and we have included additional explanations about the eligibility criteria. We have also made clear in the eligibility criteria that qualitative data from mixed-methods research were eligible for inclusion in the review. 

It is unconventional to summarise the eligibility criteria in a table. The eligibility criteria are an important part of the review and should be described in detail in the text.

We have summarised the key aspects of the eligibility criteria in the text and this is supplemented by additional detail in Table 1 which will be adjacent to the text. We have included additional detail in text to provide justification for some items within the table. We are happy to be guided by the editor about whether there is scope to increase the length of the Methods section by importing the more detailed content of this table into the main body of the text. 

Are the authors able to share the data extraction form they used or mention the aspects the extracted?

The data extraction form is presented in Table 2. This is now clarified within the text of the manuscript. 

Did the authors use a methodological quality appraisal tool (e.g. CASP or QARI)? What is currently described in this section has to do with data richness. A ‘rich’ study is not by default of ‘good’ quality.

We did not assess quality using conventional tools such as CASP as our approach was driven by sampling studies likely to be of most utility based on the data richness scale. Quality assessments in qualitative reviews is controversial and their value is debateable (Dixon-Woods, 2006) [1]. While we acknowledge that rich data is not necessarily synonymous with good quality data we have found in previous qualitative reviews that judgments about the ability of each study to contribute to the synthesis based on data thickness and richness have greater interpretative utility than judgments based on formulaic notions of study quality alone (Coventry, 2015)[2]. 

We have included additional explanation in the limitations section to support this approach. 

Authors have left a note within the text. “Findings from quantitative studies have been reported xxx to add details nearer submission depending on where we are up to with that paper].” 

This has now been updated.

It is great that the authors have included quotes from the primary studies, but it sometimes reads as though the quotes are from the authors own primary research. This is a higher-level analysis of primary research and should be presented as such.

This is an important distinction to make and we thank the reviewer for giving us a chance to rephrase reporting to ensure readers recognise that the findings are about the synthesis rather than just about the primary studies.

“The review provides novel insights and understanding about factors influencing self-management in people with SMI, which can inform critical thinking about how best to design tailored self-management interventions for this population.” This needs to be explained in the context of existing literature. 

We accept that this statement might be a little premature and vague. We have instead highlighted that a strength of our review is that it fits with the early and exploratory phases of programmatic work to design complex interventions (e.g. MRC Framework; Science of Behaviour Change). 

Supporting references

1. Dixon-Woods M, Cavers D, Agarwal S, Annandale E, Arthur A, Harvey J, et al. Conducting a critical interpretive synthesis of the literature on access to healthcare by vulnerable groups. BMC medical research methodology. 2006;6(1):35. doi: 10.1186/1471-2288-6-35. PubMed PMID: 16872487; PubMed Central PMCID: PMCPMC1559637.

2. Coventry PA, Small N, Panagioti M, Adeyemi I, Bee P. Living with complexity; marshalling resources: a systematic review and qualitative meta-synthesis of lived experience of mental and physical multimorbidity. BMC family practice. 2015;16(1):171. doi: 10.1186/s12875-015-0345-3. PubMed PMID: 26597934; PubMed Central PMCID: PMCPMC4657350.

We appreciate the time and effort that was involved in providing such a detailed response, and we look forward to hearing back from you.

Best wishes,

Dr Claire Carswell

---

## [Decision Letter · Decision Letter 1]

11 Oct 2021

Exploring the facilitators, barriers, and strategies for self-management in adults living with severe mental illness, with and without long-term conditions: a qualitative evidence synthesis

PONE-D-21-18104R1

Dear Claire Carswell,

We’re pleased to inform you that your manuscript has been judged scientifically suitable for publication and will be formally accepted for publication once it meets all outstanding technical requirements.

Kind regards,

Eleanor Ochodo

Academic Editor

PLOS ONE

Reviewers' comments:

Reviewer's Responses to Questions

**Comments to the Author**

1. If the authors have adequately addressed your comments raised in a previous round of review and you feel that this manuscript is now acceptable for publication, you may indicate that here to bypass the “Comments to the Author” section, enter your conflict of interest statement in the “Confidential to Editor” section, and submit your "Accept" recommendation.

Reviewer #1: All comments have been addressed

Reviewer #2: All comments have been addressed

2. Is the manuscript technically sound, and do the data support the conclusions?

Reviewer #1: Yes

Reviewer #2: Partly

3. Has the statistical analysis been performed appropriately and rigorously? 

Reviewer #1: N/A

Reviewer #2: N/A

4. Have the authors made all data underlying the findings in their manuscript fully available?

Reviewer #1: Yes

Reviewer #2: Yes

5. Is the manuscript presented in an intelligible fashion and written in standard English?

Reviewer #1: Yes

Reviewer #2: Yes

6. Review Comments to the Author

Reviewer #1: (No Response)

Reviewer #2: (No Response)

7. PLOS authors have the option to publish the peer review history of their article (what does this mean?). If published, this will include your full peer review and any attached files.

Reviewer #1: No

Reviewer #2: No

---

## [Editor Report · Acceptance letter]

15 Oct 2021

PONE-D-21-18104R1 

Exploring the facilitators, barriers, and strategies for self-management in adults living with severe mental illness, with and without long-term conditions: a qualitative evidence synthesis 

Dear Dr. Carswell:

I'm pleased to inform you that your manuscript has been deemed suitable for publication in PLOS ONE. Congratulations! Your manuscript is now with our production department. 

Kind regards, 

on behalf of

Prof Eleanor Ochodo 

Academic Editor

PLOS ONE